# Clinical evolution and medical resource utilization in adult patients with respiratory syncytial virus infection at a community hospital in Argentina

Agustin Bengolea[1][ID][☯][*], Juan I. Ruiz[2][ID][☯], Celina G. Vega[3][☯], Matias Manzotti[1][ID][‡], Nadia Zuccarino[3][‡], Lucila Rey-Ares[3][☯]

1 Servicio de Clínica Médica, Hospital Alemán, Ciudad de Buenos Aires, Argentina, 2 Department of Health Services Research, The University of Texas MD Anderson Cancer Center, Houston, Texas, United States of America, 3 Pfizer Argentina, Villa Adelina, Provincia de Bs As, Argentina

☯ These authors contributed equally to this work.
‡ MM and NZ also contributed equally to this work.
* agustinmbengolea@gmail.com

## Abstract

### Objective

To describe the hospital medical resources used in adults hospitalized with respiratory syncytial virus infection and to evaluate the association of comorbidities with resource utilization and clinical outcomes.

### Design

A retrospective cohort study was conducted using the electronic healthcare database of Hospital Alemán, Buenos Aires, Argentina. It included hospitalized patients aged 18 years or older who had a positive test for respiratory syncytial virus between September 2010 and December 2023. Data were analyzed using standard statistical methods following STROBE guidelines.

### Results

Among 72 hospitalized adults with confirmed respiratory syncytial virus infection, the mean length of hospital stay was 12.18 days (SD 12.91), and 27 (37.5%) patients required intensive care unit admission. Healthcare resource utilization was substantial, with 26% needing non-invasive ventilation and 11% requiring mechanical ventilation. Antibiotics (75%) and corticosteroids (68.05%) were commonly used, likely reflecting the severity of clinical presentation or the potential for bacterial coinfection. Cardiovascular comorbidities were significantly associated with severe disease outcomes and intensive care unit admission (OR 3.53, 95% CI: 1.00–12.54).

**Data availability statement:** All data generated or analyzed during this study are included in this published article/as Supporting information files.

**Funding:** This study was supported by Pfizer Inc. LRA, CGV, and NZ are employees of Pfizer Argentina SRL. AB is employed by Hospital Alemán, which received financial support from Pfizer Inc. for this study, including manuscript preparation. The funders provided financial support for the study and made suggestions on the study design and preparation of the manuscript. However, the funders had no role in data collection and analysis or decision to publish.

**Competing interests:** The authors have declared that no competing interests exist.

Comparative analysis with 226 influenza patients showed respiratory syncytial virus patients had longer hospital stays and higher intensive care unit admission rates.

## Conclusions

Respiratory syncytial virus infection in adults resulted in substantial medical resource utilization and significant intensive care unit admission and ventilation support requirements. Cardiovascular comorbidities might be associated with increased severity and intensive care unit admissions. The high rate of antibiotic use is noteworthy and warrants further investigation to understand prescribing patterns and optimize antimicrobial stewardship. Compared to influenza, respiratory syncytial virus appears to be associated with longer hospital stays and higher intensive care unit admissions, highlighting the need for tailored management strategies for respiratory syncytial virus in adult populations. Further research should focus on optimizing treatment protocols and preventive measures for respiratory syncytial virus.

## Introduction

Respiratory syncytial virus (RSV) is a significant pathogen known for causing respiratory tract infections, primarily in children, but is increasingly recognized as a cause of severe respiratory illness in adults, particularly those with underlying risk factors. In healthy older adults, the annual incidence of RSV ranges from 3% to 7%, increasing to 4% to 10% in high-risk groups (21 years of age or older with congestive heart failure or chronic pulmonary disease) [1]. Hospitalization rates for RSV in adults mirror those seen in other viral respiratory infections, such as Influenza A. However, it is important to note that individuals with RSV appear to experience a less favorable clinical course than others [2,3], as evidenced by a mortality rate spanning from 6% to 8% among adults 60 years of age or older who are hospitalized due to this infection [4].

In Latin America, a systematic review [5] that included eighteen studies revealed a substantial prevalence of RSV in adults suffering from respiratory infections (0% to 77.9%), influenza-like illness (1.0% to 16.4%), and community-acquired pneumonia (1.3% to 13.5%). The proportion of RSV-infected adults who were hospitalized was significantly high, ranging from 40.9% to 69.9% among those who tested positive for RSV with influenza-like illness, and reaching up to 91.7% among those with community-acquired pneumonia.

A systematic literature review [6] of forty-two studies analyzing the economic and healthcare resource utilization associated with RSV infections in adults, including different geographic locations across North America, South America, Europe, Asia, and Oceania, showed that based on economic models of RSV vaccination in adults aged ≥60 years, the current national direct medical costs of RSV infections are high. These costs were estimated to be $1.70–$3.35 billion in the US, $196.43 million in the United Kingdom, and $14.61 million in the Netherlands (adjusted to 2022 US dollars).

This burden is particularly highlighted in high-risk groups, as shown in a study by Nowalk et al. [7]. Older age and immunosuppression were identified as significant risk factors for RSV hospitalization. Adults aged 65 and older experienced hospitalization rates 7–9 times higher than those aged 18–64 (939 per 100,000 vs. 118 per 100,000). Immunosuppressed individuals faced the greatest risk, with hospitalization rates ranging from 1,288–1,562 per 100,000.

Despite the impact and relevance of RSV infection, global data in adults is scarce, particularly in Argentina [5,8]. Further studies are needed to gain a clearer understanding of RSV's impact on the adult population and to evaluate the role of new prevention strategies.

This study aimed to describe the utilization of hospital medical resources during the hospitalization of adult patients diagnosed with RSV infection and to evaluate the association of comorbidities with resource utilization and clinical outcomes. An exploratory analysis compared this cohort with hospitalized influenza patients.

## Materials and methods

### Data source and study design

We conducted a retrospective cohort study using the electronic healthcare database of the *Hospital Alemán* of Buenos Aires, Argentina. Data necessary for the study were retrieved on January 25, 2024 from the hospital's data warehouse by an automated process, utilizing the MicroStrategy tool. This retrieval included anonymized records of hospitalizations, outpatient consultations, laboratory test results, and imaging studies. The medical informatics team conducted the data extraction specifically for research purposes, maintaining anonymization throughout. The follow-up period extended from hospital admission to discharge, capturing comprehensive data on medical resource utilization and patient outcomes during the hospital stay. Follow-up concluded with either the patient's discharge or in-hospital death.

### Inclusion criteria and cohort selection

We included hospitalized patients ≥18 years of age who had a positive test for RSV between September 2010 and December 2023. Selection criteria required a positive result for RSV in the Respiratory Film Array or an antigen-positive result in a respiratory panel. Testing is usually reserved for patients presenting with severe respiratory symptoms or those requiring hospitalization. We required patients to have at least one clinical assessment (as outpatient or inpatient) at the hospital within 12 months prior to hospital admission to ensure a comprehensive record of their medical history.

### Outcomes and covariates

The primary outcome was medical resource utilization, including the length of hospital stay, oxygen therapy utilization, administration of antibiotics, corticosteroid use, need of bronchoalveolar lavage procedures, utilization of chest X-rays, computed tomography scans, and laboratory tests.

Secondary outcomes were clinical deterioration, including intensive care unit (ICU) admission, utilization of non-invasive ventilation (NIV), requirement for mechanical ventilation support (MV), and mortality.

Covariates analyzed included demographic and clinical characteristics such as age, gender, hypertension, cancer, diabetes, smoking, chronic obstructive pulmonary disease (COPD), asthma, overweight, obesity, chronic kidney failure, solid organ transplant, bone marrow transplant, liver disease, congestive heart failure, connective tissue disease, peripheral vascular disease, peptic ulcer disease, dementia, Acquired Immune Deficiency Syndrome (AIDS), myocardial infarction, cerebrovascular accident or transient ischemic attack, hemiplegia, and institutionalization. When possible, the Charlson comorbidity index [9,10] was calculated to estimate the clinical characteristics of the cohort. The Charlson index predicts the risk of death within 1 year of hospitalization for patients with specific comorbid conditions.

We grouped various clinical characteristics to enhance the precision and robustness of our analysis and to understand their collective impact as follows: 1) cardiovascular comorbidities (including hypertension, myocardial infarction,

cerebrovascular accident or transient ischemic attack, peripheral vascular disease, and congestive heart failure), 2) respiratory comorbidities (COPD and asthma), 3) immunosuppression (AIDS, solid organ transplant, bone marrow transplant, cancer, and connective tissue disease)

As an exploratory analysis, we compared medical resource use and prognosis of the RSV cohort with a cohort of patients aged 18 or older admitted to the *Hospital Alemán* with a diagnosis of influenza infection during the same period (September 2010 to December 2023). We used the same data collection strategy applied to the RSV cohort.

### Statistical analysis

We conducted the analysis using an open-source program for statistical analysis supported by the University of Amsterdam named JASP [11]. Categorical variables were presented as frequencies and percentages, while continuous variables were expressed as means with standard deviation (SD) or medians with interquartile range (IQR), as appropriate. Bivariable analysis employed t-tests or Mann-Whitney U tests for continuous variables and chi-square or Fisher's exact tests for categorical variables. All significance tests were two-sided, with a statistical significance level set at $p < 0.10$.

Upon observing statistical significance in bivariable analysis, we utilized multivariable logistic regression models to adjust for potential confounding factors and identify covariates associated with high healthcare resource utilization and severe disease outcomes. Linear regression was applied for continuous outcomes, while logistic regression was used for dichotomous outcomes. Odds ratios (OR) with 95% confidence intervals (CI) were reported.

Quality control (QC) was performed by a medical informatics expert who was not involved in the analysis. QC involved checking coding or analysis setup, output tables/figures, and summary/interpretation to ensure data consistency and accuracy. We followed the Strengthening the Reporting of Observational Studies in Epidemiology (STROBE) reporting guidelines [12]. This study was approved by the *Hospital Alemán* Institutional Review Board (IRB)/Ethics Committee (EC) (file no.12674).

No statistical analysis was performed to compare the RSV and influenza cohorts, as this was an exploratory analysis. The purpose of this comparison was to provide descriptive insights into differences in medical resource utilization and patient outcomes between these two groups. A statistical comparison would be misleading.

## Results

### Demographic and clinical characteristics

Between 2010 and 2023, a total of 2,968 patients underwent testing with Respiratory Film Array or respiratory panel antigen tests, revealing 98 positive cases of RSV. Among these, we identified 72 hospitalized adults with confirmed RSV infections. The majority occurred in 2023 (29/72) and 2022 (11/72). Baseline characteristics for all patients are presented in Table 1.

The annual breakdown of testing, RSV-positive cases, and hospitalizations has been moved to a table in the appendix for clarity and conciseness. Please refer to Table S1 in S1 File for detailed year-by-year data.

Most of the patients, 42/72 (58.3%) were female, with a mean age of 65.19 years (SD = 18.59). Comorbidities observed included hypertension (n = 40; 55.56%), cancer (n = 28; 38.89%), smoking (n = 19; 26.39%), kidney failure (n = 15; 20.83%), diabetes mellitus (n = 14; 19.44%), chronic obstructive pulmonary disease (n = 12; 16.67%), asthma (n = 10; 13.89%), and a history of myocardial infarction (n = 10; 13.89%). The median Charlson score was 4.54 (SD = 3.16).

Among the 72 hospitalized RSV patients, 45 (62.50%) received conventional inpatient care, while 27 (37.50%) required ICU admission. Additionally, 26 patients (36.1%) had a bacterial co-infection: 6 (23.1%) from blood, 12 (46.2%) from urine, 7 (26.9%) from sputum or respiratory cultures, and 1 (3.8%) from a stool sample.

### Healthcare resource utilization

Of the 72 RSV-positive hospitalized patients, the mean duration of hospital stay was 12.18 days (SD = 12.91), with a median of 8 days and an interquartile range (IQR) of 3–16 days (Table 2). Nasal oxygen therapy was required for 32 patients (44.44%), with a mean duration of 4.17 days (SD = 6.67). Antibiotics were indicated for 54 individuals (75%), with

**Table 1. Characteristics of hospitalized patients with RSV infection.**

| Characteristic | Total = 72 n (%) |
|---|---|
| Age, mean (SD) | 65.19 (18.59) |
| Age, min and max | 21 - 97 |
| Female | 42 (58.33) |
| Charlson score, mean (SD) | 4.54 (3.16) |
| Hypertension | 40 (55.56) |
| History of Cancer | 28 (38.89) |
| History of Diabetes mellitus | 14 (19.44) |
| Smoking | 19 (26.39) |
| COPD | 12 (16.67) |
| Asthma | 10 (13.89) |
| Overweight | 6 (8.33) |
| Obesity | 8 (11.11) |
| Chronic kidney failure | 15 (20.83) |
| Solid organ transplant | 5 (6.94) |
| Bone marrow transplant | 7 (9.72) |
| Liver disease | 4 (5.56) |
| Congestive heart failure | 9 (12.50) |
| Connective tissue disease | 3 (4.17) |
| Peripheral vascular disease | 4 (5.56) |
| Peptic ulcer disease | 1 (1.39) |
| Dementia | 5 (6.94) |
| AIDS | 1 (1.39) |
| Myocardial infarction | 10 (13.89) |
| Cerebrovascular accident or transient ischemic attack | 4 (5.56) |
| Hemiplegia | 1 (1.39) |
| Institutionalized | 2 (2.78) |
| Patients with at least one chronic condition | 64 (88.88) |

RSV, respiratory syncytial virus; COPD, Chronic Obstructive Pulmonary Disease; SD, Standard deviation; AIDS, acquired immune deficiency syndrome.

an average duration of use of 7.99 days (SD = 9.72), a median of 5 days, and an IQR of 0.75 to 9.25 days. Corticosteroids were administered to 49 patients (68.05%), with a mean duration of 9.06 days (SD = 10.69), a median of 7 days, and an IQR of 0–14 days. Chest X-rays were performed in 46 patients (63.88%), while computerized tomography (CT) scans were conducted for 27 patients (37.50%). Additionally, bronchoalveolar lavage was performed in 5 patients (6.94%).

### Clinical evolution and outcomes

Nineteen patients (19/72; 26.38%), required NIV with a mean duration of 2.96 days (SD = 6.39), eight patients (8/72; 11.11%) MV, with a mean duration of 2.43 days (SD = 9.52). For patients admitted to the ICU (27 patients, 37.50%), the mean ICU stay was 5.49 days (SD = 11.73). Four (5.55%) patients died during the hospitalization with RSV positive test. Severe disease, defined as the composite of NIV and ICU, was observed in 31 patients (43.10%).

We conducted univariable and multivariable analyses of all patient characteristics to explore associations with ICU admission, MV requirement, and severe disease (Table 3). Among the comorbidities analyzed, cardiovascular comorbidities showed a significant association with severe disease (p 0.009). Patients with cardiovascular comorbidities had higher

**Table 2. Medical resource utilization during hospitalization (n = 72).**

| Resources | n (%) | Mean (SD), days | Median (IQR), days | Min-Max, days |
|---|---|---|---|---|
| Duration of hospital stay | – | 12.18 (12.91) | 8 (3-16) | 1-64 |
| ICU admission | 27 (37.50) | 5.49 (11.73) | 0 (0-6.25) | 0-65 |
| MV | 8 (11.11) | 2.43 (9.52) | 0 | 0-60 |
| NIV[a] | 19 (26.38) | 2.96 (6.39) | 0 (0-1.25) | 0-35 |
| Oxygen therapy | 32 (44.44) | 4.17 (6.67) | 0 (0-8) | 1-27 |
| Corticosteroids | 49 (68.05) | 9.06 (10.69) | 7 (0-14) | 0-55 |
| Antibiotics | 54 (75) | 7.99 (9.72) | 5 (0.75-9.25) | 0-46 |
| Radiography | 46 (63.88) | – | – | – |
| Computerized tomography scan | 27 (37.50) | – | – | – |
| Bronchoalveolar lavage | 5 (6.94) | – | – | – |

SD, standard deviation; IQR, interquartile range; ICU, intensive care unit; MV, mechanical ventilation; NIV, noninvasive ventilation.

[a]Includes NIV at ICU and general hospital ward.

odds of ICU admission (OR 3.53, 95% CI: 1.00–12.54) and severe disease (OR 3.51, 95% CI: 1.04–11.80). Respiratory comorbidities and immunosuppression did not show significant associations with ICU admission, MV requirement, or severe disease.

### Influenza infection: Characteristics and resource utilization

Among the 226 patients admitted with influenza during the study period, demographic and clinical characteristics were examined and compared to RSV patients (Table 4). The mean age of influenza patients was 62.59 years (SD = 18.77), ranging from 20 to 100 years. Of these hospitalized influenza patients, 107 (47.35%) were female. Hypertension was observed in 50 cases (22.12%). Other comorbidities such as cancer, diabetes, COPD, and asthma exhibited prevalence rates ranging from 6.20% to 33.19%.

Influenza patients received antibiotics for an average duration of 8.5 days (SD = 17.78), compared to RSV patients, who received antibiotics for 7.99 days (SD = 9.72). Corticosteroids were administered to influenza patients for an average of 7.89 days (SD = 18.65), compared to RSV patients, who received them for 9.06 days (SD = 10.69).

Oxygen therapy was required in 42.04% of influenza patients (95/225) with a mean duration of 3.56 days (SD = 7.13), while 44.44% of RSV patients (32/72) required oxygen therapy for an average of 4.17 days (SD = 6.67). The average hospital stay for influenza patients was 10.85 days (SD = 19.10), whereas RSV patients had an average stay of 12.18 days (SD = 12.91).

Among influenza patients, ICU admission averaged 4.05 days (SD = 12.36), while RSV patients had a mean ICU stay of 5.49 days (SD = 11.73). NIV was required for an average of 1.66 days (SD = 4.33) in influenza patients, compared to RSV patients, who required NIV for 2.96 days (SD = 6.39). MV was required for 2.28 days (SD = 10.14) in influenza patients, while RSV patients required MV for 2.43 days (SD = 9.52). Additional details on the resource utilization of RSV and influenza patients are available in the supporting information.

### Discussion

This study describes the healthcare burden of RSV infections in hospitalized adults. Our findings demonstrate significant medical resource utilization, prolonged hospital stays, and high rates of ICU admission and ventilation support among RSV patients. These findings align with those of a systematic review conducted by Colosia et al. [2], which reported that RSV accounted for up to 12% of medically attended acute respiratory illnesses in older adults in the United States. This review also highlighted considerable variability across clinical settings and over time. Furthermore, a considerable

**Table 3. Bivariable and multivariable analysis of the covariates associated UCI, MV requirement and severe disease.**

| Comorbidity | UCI requirement | | | | MV requirement | | | | Severe disease | | | |
|---|---|---|---|---|---|---|---|---|---|---|---|---|
| | UCI (n=27) | p | OR (95 CI) | p | MV req. n=8 | p | OR (95 CI) | p | Severe disease n=31 | P | OR (95 CI) | p |
| Age, mean (SD) | 69.07 (17.16) | 0.172 | | | 66.25 (14.06) | 0.866 | | | 69.52 (17.40) | **0.091** | 1.01 (0.97-1.04) | 0.680 |
| ≥ 60 years | 21 | 0.296 | | | 6 (75) | 1 | | | 26 (83.87) | 0.131 | | |
| Charlson score, mean (SD) | 5.07 (3.39) | 0.272 | | | 4.38 (1.6) | 0.876 | | | 5.07 (3.22) | 0.22 | | |
| Female, n (%) | 15 (55.56) | 0.807 | | | 2 (25) | **0.06** | **0.18 (0.25-1.05)** | 0.059 | 17 (54.84) | 0.636 | | |
| Cancer, n (%) | 8 (29.63) | 0.318 | | | 4 (50) | 0.703 | | | 10 (32.26) | 0.341 | | |
| Hypertension, n (%) | 19 (70.37) | **0.056** | **2.08 (0.55-7.94)** | 0.28 | 6 (75) | 0.287 | | | 22 (70.97) | **0.031** | 2.39 (0.71-8.06) | 0.160 |
| Diabetes, n (%) | 7 (25.93) | 0.360 | | | 3 (37.5) | 0.18 | | | 8 (25.81) | 0.368 | | |
| Smoking, n (%) | 8 (29.63) | 0.783 | | | 1 (12.5) | 0.672 | | | 8 (25.81) | 1 | | |
| COPD, n (%) | 6 (22.22) | 0.347 | | | 0 (0) | 0.337 | | | 6 (19.35) | 0.751 | | |
| Asthma, n (%) | 5 (18.52) | 0.486 | | | 2 (25) | 0.307 | | | 5 (16.13) | 0.736 | | |
| Overweight, n (%) | 2 (7.41) | 1 | | | 0 (0) | 1 | | | 2 (6.45) | 0.69 | | |
| Obesity, n (%) | 2 (7.41) | 0.701 | | | 1 (12.5) | 1 | | | 3 (9.68) | 1 | | |
| Chronic kidney failure, n (%) | 6 (22.22) | 1 | | | 2 (25) | 0.669 | | | 8 (25.81) | 0.394 | | |
| Solid organ transplant, n (%) | 1 (3.7) | 0.644 | | | 0 (0) | 1 | | | 1 (3.23) | 0.382 | | |
| Bone marrow transplant, n (%) | 0 (0) | 0.04 | | | 0 (0) | 1 | | | 1 (3.23) | 0.227 | | |
| Liver disease, n (%) | 0 (0) | 0.29 | | | 0 (0) | 1 | | | 0 (0) | 0.129 | | |
| Congestive heart failure, n (%) | 6 (22.22) | **0.071** | **3.02 (0.66-13.88)** | **0.16\*** | 3 (37.5) | **0.056** | **6.58 (1.1-39.43)** | **0.039** | 8 (25.81) | **0.004** | **11.79 (1.34-103.64)** | **0.026** |
| Connective tissue disease, n (%) | 0 (0) | 0.287 | | | 0 (0) | 1 | | | 1 (3.23) | 1 | | |
| Peripheral vascular disease, n (%) | 3 (11.11) | 0.145 | | | 1 (12.5) | 0.382 | | | 3 (9.68) | 0.308 | | |
| Peptic ulcer disease, n (%) | 0 (0) | 1 | | | 0 (0) | 1 | | | 0 (0) | 1 | | |
| Dementia, n (%) | 3 (11.11) | 0.357 | | | 0 (0) | 1 | | | 3 (9.68) | 0.646 | | |
| AIDS, n (%) | 0 (0) | 1 | | | 0 (0) | 1 | | | 0 (0) | 1 | | |
| Institutionalized, n (%) | 0 (0) | 0.525 | | | 0 (0) | 1 | | | 0 (0) | 0.503 | | |
| Myocardial infarction, n (%) | 4 (14.81) | 1 | | | 1 (12.5) | 1 | | | 4 (12.9) | 1 | | |
| Cerebrovascular accident or transient ischemic attack, n (%) | 3 (11.11) | 0.145 | | | 0 (0) | 1 | | | 3 (9.68) | 0.308 | | |
| Hemiplegia, n (%) | 1 (3.7) | 0.375 | | | 0 (0) | 1 | | | 1 (3.23) | 0.431 | | |
| Cardiovascular comorbidities[a], n (%) | 21 (77.78) | **0.025** | **3.53 (1.00-12.54)** | **0.051** | 7 (87.5) | 0.132 | | | 24 (77.42) | **0.009** | **3.51 (1.04-11.80)** | **0.043** |
| Respiratory comorbidities[b], n (%) | 9 (33.33) | 0.429 | | | 2 (25) | 1 | | | 9 (29.03) | 1 | | |

*(Continued)*

**Table 3.** (Continued)

| Comorbidity | UCI requirement | | | | MV requirement | | | | Severe disease | | | |
| | UCI (n = 27) | p | OR (95 CI) | p | MV req. n = 8 | p | OR (95 CI) | p | Severe disease n = 31 | P | OR (95 CI) | p |
| --- | --- | --- | --- | --- | --- | --- | --- | --- | --- | --- | --- | --- |
| Overweight/Obesity, n (%) | 4 (14.81) | 0.547 | | | 1 (12.5) | 1 | | | 5 (16.13) | 0.765 | | |
| Inmunosupression[e], (%) | 9 (33.33) | **0.054** | | | 4 (50) | 1 | | | 12 (38.71) | 0.161 | | |

[a] Cardiovascular comorbidities defined by the presence of one or more of the following: hypertension, myocardial infarction, cerebrovascular accident or transient ischemic attack, peripheral vascular disease, congestive heart failure, n (%).

[b] Respiratory comorbidities defined by the presence of one or more of the following: COPD and Asthma.

[c] Immunosuppression defined by the presence of one or more of the following: AIDS, solid organ transplant, bone marrow transplant, cancer and connective tissue disease.

RSV, respiratory syncytial virus; COPD, Chronic Obstructive Pulmonary Disease; n, number; SD, Standard deviation; AIDS, acquired immune deficiency syndrome.

proportion of RSV hospitalizations required ICU admission and mechanical ventilation, with a reported mortality rate ranging from 6% to 8%.

Although we did not directly evaluate the economic impact of RSV infection in adults, the high resource use and prolonged hospitalization reflect a significant impact. This was previously addressed by other authors. In a retrospective study published by Smithgall et al in 2020 [13], they showed a high healthcare and economic burden, illustrated by the proportion of cases requiring medical attention (40.9%) and antibiotic use (23.1%). Additionally, RSV infection led to a high proportion of affected individuals or their caregivers (40.9%) missing school or work, significantly impacting low-income communities.

The high rate of antibiotic use observed in our cohort, with 75% of patients receiving antibiotics, is noteworthy. While this may partly reflect the severity of illness in patients hospitalized with RSV and the potential for bacterial co-infection, it also raises concerns regarding potential overuse. Such extensive prescribing warrants scrutiny, particularly in the context of RSV, a viral pathogen where routine antibiotic use may lack clinical justification. A recent Norwegian cohort study of 951 hospitalizations for confirmed viral respiratory tract infections, including RSV and influenza, found that 76% of patients received antibiotics despite no evidence of bacterial infection, and discontinuation of antibiotics within one day after viral diagnosis occurred in only 15.8% of cases [14]

These findings are strikingly similar to our own and reinforce the notion that empirical antibiotic use remains common, even in high-income settings with low background levels of antibiotic resistance. Bacterial co-infection was confirmed through respiratory cultures in 26.9% of patients, a prevalence consistent with findings from a systematic review [15] of bacterial co-infections in influenza, which reported a pooled prevalence of 22.5% (95% CI: 16.5–30.0). While these results highlight the role of bacterial co-infections in severe respiratory viral illnesses, they fall short of explaining the high rates of antibiotic use observed in our cohort. This discrepancy suggests that antibiotics may have been prescribed inappropriately, in line with findings from the systematic review [16], which reported that 36.9% of antibiotic prescriptions for respiratory infections in emergency department settings were unnecessary.

These findings emphasize the importance of targeted interventions to address the misuse of antibiotics in the management of respiratory viral infections. Strengthening diagnostic capabilities, particularly with rapid tests for bacterial pathogens, is essential. Moreover, implementing evidence-based guidelines, coupled with education for clinicians on judicious antibiotic prescribing, can help mitigate inappropriate use. Future research should explore the behavioral and systemic factors driving these prescribing patterns and evaluate the impact of stewardship programs in reducing misuse. By

**Table 4. Characteristics of hospitalized RSV and Influenza patients.**

| Characteristics | RSV = 72 n (%) | Influenza = 226 n (%) |
|---|---|---|
| Age, mean (SD) | 65.19 (18.59) | 62.59 (18.77) |
| Age, min and max | 21 - 97 | 20 - 100 |
| Female | 42 (58.33) | 107 (47.35) |
| Hypertension | 40 (55.56) | 50 (22.12) |
| Cancer | 28 (38.89) | 75 (33.19) |
| Diabetes | 14 (19.44) | 23 (10.18) |
| Smoking | 19 (26.39) | 14 (6.20) |
| COPD | 12 (16.67) | 14 (6.20) |
| Asthma | 10 (13.89) | 22 (9.74) |
| Overweight | 6 (8.33) | 13 (5.75) |
| Obesity | 8 (11.11) | 6 (2.66) |
| Chronic kidney failure | 15 (20.83) | 12 (5.31) |
| Solid organ transplant | 5 (6.94) | 22 (9.74) |
| Bone marrow transplant | 7 (9.72) | 11 (4.87) |
| Liver disease | 4 (5.56) | 5 (2.21) |
| Congestive heart failure | 9 (12.50) | 1 (0.44) |
| Connective tissue disease | 3 (4.17) | 2 (0.89) |
| Peripheral vascular disease | 4 (5.56) | 1 (0.44) |
| Peptic ulcer disease | 1 (1.39) | 2 (0.89) |
| Dementia | 5 (6.94) | 0 (0) |
| AIDS | 1 (1.39) | 0 (0) |
| Patients with at least one chronic condition | 64 (88.88) | 163 (72.12) |

RSV, respiratory syncytial virus; COPD, Chronic Obstructive Pulmonary Disease; SD, Standard deviation; AIDS, acquired immune deficiency syndrome.

addressing these gaps, we can optimize the management of respiratory infections while preserving the efficacy of antibiotics for patients with confirmed bacterial infections.

Information on RSV in Latin America is scarce; however, our findings align with the limited data available, suggesting that RSV infections may be a significant cause of hospitalization in the region. A systematic review [5] reported RSV-associated hospitalization rates ranging from 40.9% to 69.9% in RSV-infected adults aged ≤50 years, with up to 19.6% requiring ICU admission or mechanical ventilation. The highest hospitalization rates were observed in individuals aged ≥65 years (26.3%), with severe outcomes (18.2%) also noted in younger hospitalized patients (18–49 years).

Regarding influenza patients, their comorbidity profiles showed varying prevalence rates and vaccination status was not explored. Comparative analyses with influenza patients revealed that RSV patients had longer hospital stays (12.18 days vs. 10.85 days) and higher rates of ICU admission (37.50% vs. 28.32%). These findings indicate that RSV may lead to more severe clinical outcomes and greater resource demands compared to influenza, with longer ICU stays and more intensive care requirements. This is consistent with systematic reviews [17,18] on the economic outcomes associated with RSV, which indicated a high economic burden and medical resource utilization equivalent to or greater than that of influenza. These reviews noted that RSV patients were older, more likely to have chronic conditions, and required longer hospital stays compared to influenza patients, highlighting the substantial and underappreciated disease burden of RSV in the United States. However, our study showed higher medical resource utilization in RSV patients with similar medical characteristics to influenza patients. It is important to note that the influenza immunization status of the patients was not documented in this study.

 

In our study, patients with cardiovascular comorbidities showed an association with ICU admission and severe disease. This is consistent with a systematic review that showed that RSV patients hospitalized with pulmonary, cardiac, and/or immunodeficiency conditions are at increased risk of complications following any respiratory tract infection [19]. About 8–13% of patients with chronic lung or heart diseases suffered from RSV illness during 1–3 years of follow-up, and about 2–20% of HSCT patients suffered from at least one RSV infection during 1–5 years post-transplantation.

This study presents limitations that merit consideration. First, the study was conducted exclusively within a single healthcare center and only included hospitalized RSV patients, limiting the generalizability of the results to broader populations or different healthcare settings in Argentina. Additionally, the retrospective design of this study may introduce potential biases and hinder the establishment of causal relationships. There is also potential testing bias, as RSV may have only been assessed in patients presenting with severe respiratory illness. Furthermore, we cannot determine the exact cause of hospitalization due to the study design. The inclusion of data from the COVID-19 pandemic years may have influenced the epidemiological behavior of the virus, requiring cautious interpretation of the results. However, we believe that most patients in this study were hospitalized due to respiratory infections, as the Respiratory Film Array or antigen tests were primarily used for patients presenting with respiratory symptoms, except during the COVID-19 pandemic when testing was also done for isolation decisions. Improving systematic testing for RSV, particularly in certain populations, is expected to enhance diagnosis and reduce the current underestimation, thereby optimizing patient management, as supported by a systematic review [17,20].

Despite these limitations, the strengths of this study include the use of a robust database and the high-complexity healthcare context of Hospital Alemán. The hospital's structured data storage system ensures the integrity and accuracy of patient records, facilitating meticulous analysis of medical resource utilization and patient outcomes. The inclusion of patients with diverse clinical presentations and outcomes further enhances the study's external validity. Also, to our knowledge the information about RSV infection in Argentina is scarce, as is shown in a previous systematic review by Bardach et. al [8].

## Conclusion

RSV infection in adults' resulted in substantial medical resource utilization, with significant ICU and ventilation support requirements. Cardiovascular comorbidities might be associated with increased disease severity and ICU admissions. Compared to influenza, RSV may lead to longer hospital stays and higher ICU admission rates, underscoring the need for tailored management strategies for RSV in adult populations. The high rate of antibiotic use observed in this study is concerning and may indicate inappropriate prescribing practices, potentially driven by diagnostic uncertainty or overestimation of bacterial co-infections. Further research should focus on optimizing treatment protocols and preventive measures to better manage the burden of RSV in this vulnerable group.

## Supporting information

**S1 File. Supplementary materials including Table S1 (Annual Breakdown of RSV Testing, Positivity, and Hospitalizations), Table S2 (Resource utilization in patients hospitalized with RSV), Table S3 (Resource utilization in patients hospitalized with RSV and Influenza), and the dataset access link.**
(DOCX)

## Acknowledgments

The authors would like to thank Reiko Sato for reviewing the manuscript.

## Author contributions

**Conceptualization:** Agustin Bengolea, Juan I. Ruiz, Celina G. Vega, Nadia Zuccarino, Lucila Rey-Ares.

**Data curation:** Matias Manzotti.

**Formal analysis:** Agustin Bengolea, Lucila Rey-Ares.

**Funding acquisition:** Agustin Bengolea.

**Investigation:** Agustin Bengolea, Juan I. Ruiz, Celina G. Vega, Lucila Rey-Ares.

**Methodology:** Agustin Bengolea, Juan I. Ruiz, Celina G. Vega, Lucila Rey-Ares.

**Project administration:** Agustin Bengolea.

**Resources:** Agustin Bengolea, Celina G. Vega.

**Software:** Matias Manzotti.

**Supervision:** Agustin Bengolea, Juan I. Ruiz, Celina G. Vega, Lucila Rey-Ares.

**Validation:** Agustin Bengolea, Juan I. Ruiz, Matias Manzotti.

**Visualization:** Agustin Bengolea, Juan I. Ruiz.

**Writing – original draft:** Agustin Bengolea, Juan I. Ruiz, Lucila Rey-Ares.

**Writing – review & editing:** Agustin Bengolea, Juan I. Ruiz, Celina G. Vega, Nadia Zuccarino, Lucila Rey-Ares.

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
