## [Decision Letter · Decision Letter 0]

8 Jan 2025

PONE-D-24-49578Clinical evolution and medical resource utilization in adult patients with respiratory syncytial virus infection at a community hospital in ArgentinaPLOS ONE

Dear Dr. Bengolea,

Thank you for submitting your manuscript to PLOS ONE. After careful consideration, we feel that it has merit but does not fully meet PLOS ONE’s publication criteria as it currently stands. Therefore, we invite you to submit a revised version of the manuscript that addresses the points raised during the review process.

We look forward to receiving your revised manuscript.

Kind regards,

André Ricardo Ribas Freitas

Academic Editor

PLOS ONE

Journal Requirements:

“The research described herein was supported by Pfizer Inc. LRA, CGV, and NZ are

employed by Pfizer Argentina SRL. AB is an employee of Hospital Alemán, which

received financial support from Pfizer Inc. for this study (including manuscript

preparation).”

3. We noted in your submission details that a portion of your manuscript may have been presented or published elsewhere. [A poster presenting preliminary results from this study is scheduled for presentation at the upcoming ISPOR Europe conference in Barcelona. The data and analyses in this manuscript encompass a comprehensive, final dataset with additional insights and detailed analyses that go beyond the scope of the poster. Accordingly, this submission does not constitute dual publication, as it provides an expanded and complete version of the findings.] Please clarify whether this [conference proceeding or publication] was peer-reviewed and formally published. If this work was previously peer-reviewed and published, in the cover letter please provide the reason that this work does not constitute dual publication and should be included in the current manuscript.

4. In this instance it seems there may be acceptable restrictions in place that prevent the public sharing of your minimal data. However, in line with our goal of ensuring long-term data availability to all interested researchers, PLOS’ Data Policy states that authors cannot be the sole named individuals responsible for ensuring data access (http://journals.plos.org/plosone/s/data-availability#loc-acceptable-data-sharing-methods).

5. Please include captions for your Supporting Information files at the end of your manuscript, and update any in-text citations to match accordingly. Please see our Supporting Information guidelines for more information: http://journals.plos.org/plosone/s/supporting-information .

Additional Editor Comments:

Dear Dr. Agustin Bengolea,

Thank you for submitting your manuscript titled "Clinical evolution and medical resource utilization in adult patients with respiratory syncytial virus infection at a community hospital in Argentina" to PLOS ONE.

After review by three experts, all have recommended minor revisions. We appreciate the significant contribution of your work and kindly request that you prepare a revised version addressing the reviewers' comments and suggestions. Please include a detailed response letter explaining how each point has been addressed.

If you require any clarification during the revision process, feel free to reach out to me for assistance.

Best regards,

André Ricardo Ribas Freitas

Academic Editor

PLOS ONE

Reviewers' comments:

Reviewer's Responses to Questions

**Comments to the Author**

1. Is the manuscript technically sound, and do the data support the conclusions?

Reviewer #1: Yes

Reviewer #2: Partly

Reviewer #3: Yes

2. Has the statistical analysis been performed appropriately and rigorously? 

Reviewer #1: Yes

Reviewer #2: Yes

Reviewer #3: Yes

3. Have the authors made all data underlying the findings in their manuscript fully available?

Reviewer #1: Yes

Reviewer #2: Yes

Reviewer #3: Yes

4. Is the manuscript presented in an intelligible fashion and written in standard English?

Reviewer #1: Yes

Reviewer #2: Yes

Reviewer #3: Yes

5. Review Comments to the Author

Reviewer #1: This article presents data on the RSV burden in adults from a single center in Argentina. Although it adds little to the existing body of evidence on the topic, it is well-written.

A few minor comments.

Table 1: add the % of patients with at least one chronic condition

For influenza patients: Add the % of CTC and ATB use. Add if available % of flu vaccination.

Would it be possible to add information on co-infections for the RSV patients?

I am surprised by the high rate of Antibiotic prescriptions in these patients. (Is this due to bacterial co-infection? Or misuse ?) Can you comment on this rate and the high CTC rate as well?

Reviewer #2: The authors describe differences between the two cohorts (syncytial virus and influenza) and conclude with some differences between them. ¿Are those differences significant? ¿It would be appropriate to explain why a statistical study was not carried out between the two cohorts?.

The authors report that 27 patients required ICUs; nineteen received NIV and 8 received mechanical ventilation. Then, they say that 31 patients met the criterion of "severe disease", which includes the compound of non-invasive ventilation and admission to the ICU. Why that difference of 4 patients?

Reviewer #3: This is an important study on RSV in Argentina, for which there is little published evidence, especially regarding the burden of disease among adults and the healthcare utilization.

The authors conducted a retrospective cohort study for which they identified all RSV positive cases in a 14 year period in a single Argentinian center. They also report the number of all patients that were subjected to viral testing. The manuscript is well structured and written and with an easy to follow, logical chain of thoughts. I will give comments in chronological order and will highlight aspects that I deem critical that should be revised to improve the quality of the work.

Of note, the study was financially supported by Pfizer, and three of the authors are employed by the company.

1. Title: no comments

2. Abstract: I think the finding that 75% were treated with antibiotics is absolutely worrying. This should be picked up in the conclusions section as well, depending on further investigations/analyses that I comment on in the results part.

3. Introduction: no comments

4. Methods: please briefly explain what the algorithm is for microbiological/viral testing, as the overall number that you report in the results for such a long period of time is rather low (212 per year?)

5. Results: please provide numbers also per year, so potential time trends, both with regard to total tests ordered and positive tests, can be assessed

6. Results: I think one crucial information that is lacking at the moment is information on the reasoning of antibiotic treatment. It could be helpful to include data on microbiological diagnostics in these cases (they are not that many, so it should be feasible), to approximate whether there is a reasonable number of patients with a bacterial co-infection. I presume that anyhow the vast majority will be unneccesary overtreatment of viral cases.

7. Discussion: The issue of overtreatment could be picked up even more here

6. PLOS authors have the option to publish the peer review history of their article (what does this mean? ). If published, this will include your full peer review and any attached files.

**Do you want your identity to be public for this peer review?** For information about this choice, including consent withdrawal, please see our Privacy Policy .

Reviewer #1: **Yes: ** Paul Loubet

Reviewer #2: No

Reviewer #3: No

---

## [Author Response · Author response to Decision Letter 0]

16 Feb 2025

Dear Editor,

We thank you for the valuable comments and constructive feedback that have significantly improved this manuscript. Below, we address each comment in detail.

Journal Requirements:

Addressed

2. Thank you for stating the following financial disclosure: “The research described herein was supported by Pfizer Inc. LRA, CGV, and NZ are employed by Pfizer Argentina SRL. AB is an employee of Hospital Alemán, which received financial support from Pfizer Inc. for this study (including manuscript preparation).” Please state what role the funders took in the study. If the funders had no role, please state: "The funders had no role in study design, data collection and analysis, decision to publish, or preparation of the manuscript." If this statement is not correct you must amend it as needed. Please include this amended Role of Funder statement in your cover letter; we will change the online submission form on your behalf.

Included in the cover letter

3. We noted in your submission details that a portion of your manuscript may have been presented or published elsewhere. [A poster presenting preliminary results from this study is scheduled for presentation at the upcoming ISPOR Europe conference in Barcelona. The data and analyses in this manuscript encompass a comprehensive, final dataset with additional insights and detailed analyses that go beyond the scope of the poster. Accordingly, this submission does not constitute dual publication, as it provides an expanded and complete version of the findings.] Please clarify whether this [conference proceeding or publication] was peer-reviewed and formally published. If this work was previously peer-reviewed and published, in the cover letter please provide the reason that this work does not constitute dual publication and should be included in the current manuscript.

Included in the cover letter

4. In this instance it seems there may be acceptable restrictions in place that prevent the public sharing of your minimal data. However, in line with our goal of ensuring long-term data availability to all interested researchers, PLOS’ Data Policy states that authors cannot be the sole named individuals responsible for ensuring data access (http://journals.plos.org/plosone/s/data-availability#loc-acceptable-data-sharing-methods). Data requests to a non-author institutional point of contact, such as a data access or ethics committee, helps guarantee long term stability and availability of data. Providing interested researchers with a durable point of contact ensures data will be accessible even if an author changes email addresses, institutions, or becomes unavailable to answer requests. Before we proceed with your manuscript, please also provide non-author contact information (phone/email/hyperlink) for a data access committee, ethics committee, or other institutional body to which data requests may be sent. If no institutional body is available to respond to requests for your minimal data, please consider if there any institutional representatives who did not collaborate in the study, and are not listed as authors on the manuscript, who would be able to hold the data and respond to external requests for data access? If so, please provide their contact information (i.e., email address). Please also provide details on how you will ensure persistent or long-term data storage and availability.

Included in the cover letter

Included in the manuscript

All references were reviewed and no cited papers have been retracted

Reviewers' comments:

Reviewer #1: This article presents data on the RSV burden in adults from a single center in Argentina. Although it adds little to the existing body of evidence on the topic, it is well-written. A few minor comments.

1. Table 1: add the % of patients with at least one chronic condition

For influenza patients: Add the % of CTC and ATB use. Add if available % of flu vaccination.

The percentages of corticosteroid and antibiotic use for influenza and RSV are available in the supplementary information. Unfortunately, we do not have data on flu vaccination rates, as this information was not recorded in the medical histories.

2. Would it be possible to add information on co-infections for the RSV patients?

Yes, we have included them in the results section. The relevant text states: “Among the 72 hospitalized RSV patients, 45 (62.5%) received conventional inpatient care, while 27 (37.5%) required ICU admission. Additionally, 26 patients (36.1%) had a bacterial co-infection: 6 (23.1%) from blood, 12 (46.2%) from urine, 7 (26.9%) from sputum or respiratory cultures, and 1 (3.8%) from a stool sample.”

3. I am surprised by the high rate of Antibiotic prescriptions in these patients. (Is this due to bacterial co-infection? Or misuse ?) Can you comment on this rate and the high CTC rate as well?

Included in the discussion section: “The high rate of antibiotic use observed in our cohort, with 75% of patients receiving antibiotics, is noteworthy. While this may partly reflect the severity of illness in patients hospitalized with RSV and the potential for bacterial co-infection, it also raises concerns regarding potential overuse. Such extensive prescribing warrants scrutiny, particularly in the context of RSV, a predominantly viral pathogen where routine antibiotic use may lack clinical justification.”

Reviewer #2:

1. The authors describe differences between the two cohorts (syncytial virus and influenza) and conclude with some differences between them. ¿Are those differences significant? ¿It would be appropriate to explain why a statistical study was not carried out between the two cohorts?.

No statistical analysis was conducted to compare the RSV and influenza cohorts, as this was an exploratory analysis. This has been clarified in the methods section: “No statistical analysis was performed to compare the RSV and influenza cohorts, as this was an exploratory analysis. The purpose of this comparison was to provide descriptive insights into differences in medical resource utilization and patient outcomes between these two groups. A statistical comparison would be misleading”

2. The authors report that 27 patients required ICUs; nineteen received NIV and 8 received mechanical ventilation. Then, they say that 31 patients met the criterion of "severe disease", which includes the compound of non-invasive ventilation and admission to the ICU. Why that difference of 4 patients?

The difference arises because not all patients who received non-invasive ventilation (NIV) were in the ICU. At the Hospital Alemán, NIV can be provided outside the ICU setting, which accounts for the discrepancy in the numbers.

Reviewer #3:

This is an important study on RSV in Argentina, for which there is little published evidence, especially regarding the burden of disease among adults and the healthcare utilization.

The authors conducted a retrospective cohort study for which they identified all RSV positive cases in a 14 year period in a single Argentinian center. They also report the number of all patients that were subjected to viral testing. The manuscript is well structured and written and with an easy to follow, logical chain of thoughts. I will give comments in chronological order and will highlight aspects that I deem critical that should be revised to improve the quality of the work. Of note, the study was financially supported by Pfizer, and three of the authors are employed by the company.

1. Title: no comments

2. Abstract: I think the finding that 75% were treated with antibiotics is absolutely worrying. This should be picked up in the conclusions section as well, depending on further investigations/analyses that I comment on in the results part.

We have included in the manuscript and the abstract: “Antibiotics (75%) and corticosteroids (68.05%) were commonly used, likely reflecting the severity of clinical presentation or the potential for bacterial coinfection.” “The high rate of antibiotic use observed in our cohort, with 75% of patients receiving antibiotics, is noteworthy. While this may partly reflect the severity of illness in patients hospitalized with RSV and the potential for bacterial co-infection, it also raises concerns regarding potential overuse. Such extensive prescribing warrants scrutiny, particularly in the context of RSV, a predominantly viral pathogen where routine antibiotic use may lack clinical justification.”

3. Introduction: no comments

4. Methods: please briefly explain what the algorithm is for microbiological/viral testing, as the overall number that you report in the results for such a long period of time is rather low (212 per year?)

No algorithm for microbiological/viral testing is applied at the Hospital Alemán. Testing is typically reserved for patients with severe respiratory symptoms or those requiring hospitalization. This has been clarified in the methods section: “Testing is usually reserved for patients presenting with severe respiratory symptoms or those requiring hospitalization.”

5. Results: please provide numbers also per year, so potential time trends, both with regard to total tests ordered and positive tests, can be assessed

Included in the results and in the supplementary information section: “The annual breakdown of testing, RSV-positive cases, and hospitalizations has been moved to a table in the appendix for clarity and conciseness. Please refer to Table S1 in the for detailed year-by-year data.

Table S1.

6. Results: I think one crucial information that is lacking at the moment is information on the reasoning of antibiotic treatment. It could be helpful to include data on microbiological diagnostics in these cases (they are not that many, so it should be feasible), to approximate whether there is a reasonable number of patients with a bacterial co-infection. I presume that anyhow the vast majority will be unneccesary overtreatment of viral cases.

Included in the results section: “Among the 72 hospitalized RSV patients, 45 (62.50%) received conventional inpatient care, while 27 (37.50%) required ICU admission. Additionally, 27 patients (37.5%) had a bacterial co-infection.”

Table S1. Annual Breakdown of RSV Testing, Positivity, and Hospitalizations

Year RSV Positivity (Positive Cases / Total Tests) Hospitalization Rate (Hospitalizations / RSV Positive Cases)

2010 0.00% (0 / 3) 0.00% (0 / 0)

2011 25.00% (1 / 4) 100.00% (1 / 1)

2012 0.00% (0 / 14) 0.00% (0 / 0)

2013 4.35% (1 / 23) 100.00% (1 / 1)

2014 9.68% (6 / 62) 66.67% (4 / 6)

2015 1.95% (3 / 154) 66.67% (2 / 3)

2016 1.72% (4 / 232) 75.00% (3 / 4)

2017 1.98% (5 / 252) 60.00% (3 / 5)

2018 1.66% (4 / 241) 25.00% (1 / 4)

2019 2.57% (12 / 467) 50.00% (6 / 12)

2020 0.69% (2 / 290) 100.00% (2 / 2)

2021 6.54% (10 / 153) 90.00% (9 / 10)

2022 3.14% (13 / 414) 84.62% (11 / 13)

2023 5.61% (37 / 659) 78.38% (29 / 37)

Total 3.30% (98 / 2968) 73.47% (72 / 98)

RSV, respiratory syncytial virus

7. Discussion: The issue of overtreatment could be picked up even more here

Included in the discussion section: “The high rate of antibiotic use observed in our cohort, with 75% of patients receiving antibiotics, is noteworthy. While this may partly reflect the severity of illness in patients hospitalized with RSV and the potential for bacterial co-infection, it also raises concerns regarding potential overuse. Such extensive prescribing warrants scrutiny, particularly in the context of RSV, a predominantly viral pathogen where routine antibiotic use may lack clinical justification.

Bacterial co-infection was confirmed through respiratory cultures in 26.9% of patients, a prevalence consistent with findings from a systematic review[14] of bacterial co-infections in influenza, which reported a pooled prevalence of 22.5% (95% CI: 16.5–30.0). While these results highlight the role of bacterial co-infections in severe respiratory viral illnesses, they fall short of explaining the high rates of antibiotic use observed in our cohort. This discrepancy suggests that antibiotics may have been prescribed inappropriately, in line with findings from the systematic review[15], which reported that 36.9% of antibiotic prescriptions for respiratory infections in emergency department settings were unnecessary.

These findings emphasize the importance of targeted interventions to address the misuse of antibiotics in the management of respiratory viral infections. Strengthening diagnostic capabilities, particularly with rapid tests for bacterial pathogens, is essential. Moreover, implementing evidence-based guidelines, coupled with education for clinicians on judicious antibiotic prescribing, can help mitigate inappropriate use. Future research should explore the behavioral and systemic factors driving these prescribing patterns and evaluate the impact of stewardship programs in reducing misuse. By addressing these gaps, we can optimize the management of respiratory infections while preserving the efficacy of antibiotics for patients with confirmed bacterial infections..”

---

## [Decision Letter · Decision Letter 1]

31 Mar 2025

PONE-D-24-49578R1Clinical evolution and medical resource utilization in adult patients with respiratory syncytial virus infection at a community hospital in ArgentinaPLOS ONE

Dear Dr. Bengolea,

Thank you for submitting your manuscript to PLOS ONE. After careful consideration, we feel that it has merit but does not fully meet PLOS ONE’s publication criteria as it currently stands. Therefore, we invite you to submit a revised version of the manuscript that addresses the points raised during the review process.

We look forward to receiving your revised manuscript.

Kind regards,

André Ricardo Ribas Freitas

Academic Editor

PLOS ONE

Journal Requirements:

Additional Editor Comments :

Dear Dr. Bengolea,

Thank you for your revised submission of the manuscript titled "Clinical evolution and medical resource utilization in adult patients with respiratory syncytial virus infection at a community hospital in Argentina" (Manuscript ID: PONE-D-24-49578R1).

After reviewing the updated version and considering the feedback from both reviewers, I recommend that you proceed with a minor revision. The reviewers noted that you have addressed the majority of the concerns raised previously. However, two points still require your attention:

Table 2 – Please clarify the number of patients requiring NIV, ensuring consistency with your response and providing the adjusted mean days accordingly (Reviewer 2).

Antibiotic use – Reviewer 3 suggested further contextualization of the observed antibiotic use rate, including a comparison with other studies, and also flagged a phrase that should be clarified ("a predominantly viral pathogen").

Please revise the manuscript accordingly and submit a point-by-point response to the reviewers’ comments.

We look forward to receiving your revised manuscript.

Best regards,

André Ricardo Ribas Freitas

Academic Editor

PLOS ONE

Reviewers' comments:

Reviewer's Responses to Questions

**Comments to the Author**

1. If the authors have adequately addressed your comments raised in a previous round of review and you feel that this manuscript is now acceptable for publication, you may indicate that here to bypass the “Comments to the Author” section, enter your conflict of interest statement in the “Confidential to Editor” section, and submit your "Accept" recommendation.

Reviewer #2: All comments have been addressed

Reviewer #3: (No Response)

2. Is the manuscript technically sound, and do the data support the conclusions?

Reviewer #2: Yes

Reviewer #3: Yes

3. Has the statistical analysis been performed appropriately and rigorously? 

Reviewer #2: Yes

Reviewer #3: Yes

4. Have the authors made all data underlying the findings in their manuscript fully available?

Reviewer #2: Yes

Reviewer #3: Yes

5. Is the manuscript presented in an intelligible fashion and written in standard English?

Reviewer #2: Yes

Reviewer #3: Yes

6. Review Comments to the Author

Reviewer #2: In Table2, the number of patients recquired NIV were 19, but in your response 4 received NIV out of the ICU

Thus ,the number of NIV in Table 2 must be 23 , with adjusted mean days (SD)

Reviewer #3: Thank you for addressing most of my concerns.

One remaining issue is the antibiotic overtreatment. I think it still lacks contextualization. Is the rate observed in your cohort high, low or average? can you cite some studies for a better comparison?

Furthermore, what do you mean with "a predominantly viral pathogen"??? RSV is a virus, period.

7. PLOS authors have the option to publish the peer review history of their article (what does this mean? ). If published, this will include your full peer review and any attached files.

**Do you want your identity to be public for this peer review?** For information about this choice, including consent withdrawal, please see our Privacy Policy .

Reviewer #2: No

Reviewer #3: No

---

## [Author Response · Author response to Decision Letter 1]

2 Apr 2025

Dear Reviewers,

We sincerely thank you for your valuable comments and constructive feedback, which have contributed to significantly improving the quality of our manuscript. Below, we provide detailed responses to each of your points.

Reviewer #2:

1. In Table2, the number of patients recquired NIV were 19, but in your response 4 received NIV out of the ICU. Thus ,the number of NIV in Table 2 must be 23 , with adjusted mean days (SD)

Response to reviewers:

Table 2 summarizes the results for the entire study population, including both patients who required ICU admission and those who did not. As previously mentioned, at Hospital Alemán, non-invasive ventilation (NIV) can be administered not only in the ICU but also in general wards.

As stated in the manuscript: “Nineteen patients (19/72; 26.38%) required NIV, with a mean duration of 2.96 days (SD = 6.39); eight patients (8/72; 11.11%) required mechanical ventilation (MV), with a mean duration of 2.43 days (SD = 9.52). Among the 27 patients (37.50%) admitted to the ICU, the mean ICU stay was 5.49 days (SD = 11.73). Four patients (5.55%) with RSV-positive tests died during hospitalization. Severe disease, defined as the composite of NIV and ICU admission, was observed in 31 patients (43.10%).”

In our previous response, we stated: “The difference arises because not all patients who received non-invasive ventilation (NIV) were in the ICU. At Hospital Alemán, NIV can be provided outside the ICU setting, which accounts for the discrepancy in the numbers.”

We now realize that our explanation may have led to some misunderstanding, and we thank the reviewer for pointing this out. To clarify, we have added the following footnote for the NIV to Table 2: “Includes NIV at ICU and general hospital ward”

Reviewer #3:

1. One remaining issue is the antibiotic overtreatment. I think it still lacks contextualization. Is the rate observed in your cohort high, low or average? can you cite some studies for a better comparison?

We have included the following in the discussion section: “A recent Norwegian cohort study of 951 hospitalizations for confirmed viral respiratory tract infections, including RSV and influenza, found that 76% of patients received antibiotics despite no evidence of bacterial infection, and discontinuation of antibiotics within one day after viral diagnosis occurred in only 15.8% of cases[14]

These findings are strikingly similar to our own and reinforce the notion that empirical antibiotic use remains common, even in high-income settings with low background levels of antibiotic resistance.”

2. Furthermore, what do you mean with "a predominantly viral pathogen"??? RSV is a virus, period.

Thank you for pointing this out. You are correct—referring to RSV as a "predominantly viral pathogen" was inaccurate. We have corrected the sentence to: “Such extensive prescribing warrants scrutiny, particularly in the context of RSV, a viral pathogen for which routine antibiotic use may lack clinical justification.”

Kind regards,

Agustin Bengolea

---

## [Editor Report · Decision Letter 2]

30 Apr 2025

Clinical evolution and medical resource utilization in adult patients with respiratory syncytial virus infection at a community hospital in Argentina

PONE-D-24-49578R2

Dear Dr. Bengolea,

We’re pleased to inform you that your manuscript has been judged scientifically suitable for publication and will be formally accepted for publication once it meets all outstanding technical requirements.

Kind regards,

André Ricardo Ribas Freitas

Academic Editor

PLOS ONE

Additional Editor Comments (optional):

Dear Dr. Bengolea and colleagues,

Thank you for submitting the revised version of your manuscript entitled “Clinical evolution and medical resource utilization in adult patients with respiratory syncytial virus infection at a community hospital in Argentina” (Manuscript ID: PONE-D-24-49578R2).

I have reviewed your responses to the reviewers' comments and the updated manuscript. I am pleased to inform you that your revisions have successfully addressed all remaining concerns, and I am happy to accept your manuscript for publication in PLOS ONE.

Congratulations on your important contribution. Your work adds valuable insights into the clinical burden and healthcare impact of RSV infections in adults in real-world settings, and I believe it will be of interest to many readers.

I wish you all the best with the final steps of publication and in your ongoing research.

Warm regards,

André Ricardo Ribas Freitas, MD, PhD
---

## [Editor Report · Acceptance letter]

PONE-D-24-49578R2

PLOS ONE

Dear Dr. Bengolea,

I'm pleased to inform you that your manuscript has been deemed suitable for publication in PLOS ONE. Congratulations! Your manuscript is now being handed over to our production team.

Kind regards,

on behalf of

Dr. André Ricardo Ribas Freitas

Academic Editor

PLOS ONE